# Highly Hydrophobic, Homogeneous Suspension and Resin by Graft Copolymerization Modification of Cellulose Nanocrystal (CNC)

**Zhuofan Xu [1], Shuting Peng [1], Guofu Zhou [1,2,3,4] and Xuezhu Xu [1,2,*]**

[1] Guangdong Provincial Key Laboratory of Optical Information Materials and Technology and Institute of Electronic Paper Displays, South China Academy of Advanced Optoelectronics, South China Normal University, Guangzhou 510006, China; zhuofanxu@m.scnu.edu.cn (Z.X.); stpeng@m.scnu.edu.cn (S.P.); guofu.zhou@m.scnu.edu.cn (G.Z.)

[2] National Center for International Research on Green Optoelectronics, South China Normal University, Guangzhou 510006, China

[3] Shenzhen Guohua Optoelectronics Tech. Co. Ltd., Shenzhen 518110, China

[4] Academy of Shenzhen Guohua Optoelectronics, Shenzhen 518110, China

* Correspondence: xxzxu@m.scnu.edu.cn

**Abstract:** Cellulose nanocrystal (CNC) is a nanoscale colloid with superior potential for coatings, liquid crystal displays, and optoelectronics. However, to date, the presence of hydrophilicity still limits its application. Multifunction via graft copolymerization modification of CNC appears to be breaking into a new direction. In this study, we used the residual hydroxyl groups on the CNC to react with 2-bromoisobu-tyryl bromide, and the initiator was therefore anchored on the CNC surface. Through atom transfer radical polymerization (ATRP), CNC was successfully grafted to azobenzene monomer, i.e., 9-[4-[2-[4-(trifluorometh) phenyl] diazenyl] phenoxy] nonayl acrylate (FAZO). After a series of characterization methods, such as FTIR, NMR and XRD, it was found that the surface water contact angle of the CNC-PFAZO prepared by the modification was as high as 134.4°, and the high hydrophilicity of this material could be maintained for up to one month, even longer.

**Keywords:** cellulose nanocrystal; azobenzene; hydrophobic

## 1. Introduction

As a cellulose-based material, cellulose nanocrystals (CNC) have the advantages of being widely sourced and renewable. What is fascinating is that CNC have excellent mechanical and optical properties, which endows it with a wide range of application scenarios in nanocomposites, and thus can obtain nanocomposites with high added value [1,2]. Up to now, there have been a large number of research reports on cellulose-based composite materials. Specifically, CNC surface modification generally has two forms: covalent bond modification and non-covalent bond modification. Shikha Shrestha et al. [3] explored a covalent surface-modified cellulose nanofiber-epoxy nanocomposite system. They mentioned that, compared with other cellulose-based fillers, the surface modification of TEMPO-oxidized CNFs could improve the reinforcement effect of nanocomposites and enhance the final performance even at low filler content. Chowdhury et al. [4] studied nanocomposite coatings formed by cross-linking cellulose nanocrystals and water-based blocked polyisocyanate at different curing temperatures, where CNC and polyisocyanate (PCI) are mechanically mixed into the same situation. The conclusion is that after CNC and PIC are cross-linked, the hydrophilicity is significantly reduced by 3 times.

However, a key point is that the CNC surface contains a large number of hydroxyl groups, which gives CNC a high degree of hydrophilicity and limits the application of CNC in common nonpolar solvents and hydrophobic polymer matrixes. Fortunately, a large number of hydroxyl groups are then available to facilitate subsequent oxidation, esterification and grafting for surface covalent modification, which helps the design and application of cellulose-based nanocomposites [5–7]. In order to achieve the purpose of increasing the application of CNC in the hydrophobic polymer matrix, polypropylene, polypropylene, polytetrahydrofuran, polycaprolactone, polyethylene glycol are commonly used to modify the surface of the CNC. However, the existence of steric hindrance significantly limits the graft density on the surface of the CNC [8–10]. It is worth mentioning that "grafting from" is one of the common methods to obtain high graft density. Atom transfer radical polymerization (ATRP), as a typical "grafting from" method, is by far the most popular method, focusing on ring-opening polymerization, but it is affected by the grafting method and the stability of the grafted nanocellulose colloid [11–13], and the water contact angle can be increased to the range of 62°–94°. However, it should be noted that it is easy to encounter the pre-attachment of initiator, polymerization reactions themselves, and postprocessing/separation very lengthy problems in the ATRP process. In addition, Morandi et al. [14] demonstrated the possibility of changing the length of the graft by changing the initial ratio of monomer to the initiator and the final conversion rate.

Azobenzene compounds are one of the most diverse organic dyes, and there are two isomers, cis and trans. Its photochromic properties are based on the cis-trans isomerism of N==N bonds. In the reversible photochemical reaction process, the configuration change of the molecule will cause the color change [15–17]. Therefore, this type of material can be used for materials such as reversible optical data storage and photolithographic diffraction gratings. In addition, in order to improve the stability performance of the material and improve the processing performance of the material, the host/guest polymer material can be formed by dispersing (dissolving) the host of small azobenzene molecules in the polymer guest. Alternatively, small azobenzene molecules can be covalently bonded to the polymer molecular chain [18]. It is worth mentioning that there have been related reports on the synthesis and research of cellulose-based azobenzene polymers [19]. Stannett et al. [20] used a photochemical induction method to graft 4-(N-ethyl-N-2-acryloyloxyethyl)amino-4′-nitro-azobenzene onto cellulose. Tripathy's group [21] combined 4-cyanophenyl azophenol with ultra-high molecular weight cellulose through the Mitsunobu reaction to synthesize an azo cellulose polymer. Zheng et al. [22] obtained the supramolecular complex of azocellulose (AZO-HPMC) by esterifying the hydroxyl group on HPMC with the acid chloride group on 4-phenylazobenzoyl chloride.

In this study, we aimed to increase the processibility of CNC suspension. For example, if we can make the azo-polymer grafted CNC suspension that can well suspend in an organic solvent, the azo-polymer CNC can well coat on any surfaces showing wanted color as well so that it will find great use in many areas. Herein, we designed, successfully grafted cellulose nanocrystals, and functionalized with α-bromoisobutyryl bromide initiating sites for ATRP of azobenzene monomer, i.e., 9-[4-[2-[4-(trifluorometh) phenyl] diazenyl] phenoxy] nonayl acrylate (FAZO). We reported a stepwise graft copolymerization modification of CNC and successfully obtained a homogeneous and yellow-colored FAZO-grafted CNC suspension well dispersed in an organic solvent. The reaction is facile and will benefit the colloidal and interfacial design, and we trust it will find further applications in many areas like responsive wettability, optical information storage, optical switching.

## 2. Results and Discussion

The synthetic route of FAZO monomer is outlined in Figure 1 (See the support information for more experimental details). IR spectra of the FAZOH and FAZ9OH displays the characteristic phenolic hydroxyl group stretch at 3328 cm$^{-1}$, and the C=C bond in the benzene ring stretch at 1595 cm$^{-1}$. Moreover, the absorption peaks at 1325 cm$^{-1}$, 1257 cm$^{-1}$, 1140 cm$^{-1}$, and 1083 cm$^{-1}$ were the absorption peaks of the C–F bond in trifluoromethyl, indicating that phenol and 4-trifluoromethyl aniline had the diazo-coupling reaction, and 4-trifluoromethyl-4′-azophenol (FAZOH)

was obtained. 4-trifluoromethyl4′-(9-hydroxy-nonoxy) azobenzene was synthesized through the 4-trifluoromethyl4′-azophenol etherified with 9-bromo-1-nonanol (FAZ9OH). This can be seen from the difference between the IR spectra of FAZOH and FAZ9OH. IR spectra of the FAZ9OH displays the characteristic methylene stretch at 2949 cm$^{-1}$, 2848 cm$^{-1}$, which indicates FAZ9OH was successfully synthesized.

**Figure 1.** Synthesis of (**a**) 9-[4-[2-[4-(trifluorometh) phenyl] diazenyl] phenoxy] nonayl acrylate (FAZO)-monomers and (**b**) poly-FAZO brushes on cellulose crystals.

Under alkaline conditions, only an O-alkylation reaction of phenol oxygen anions occurs. In alkaline conditions, phenoxy anion is a good nucleophile, but the alkylation reaction of benzene ring must first destroy the aromatic system of the benzene ring, and the reaction energy is greatly increased, making O-alkylation the reaction is faster than C-alkylation and is irreversible. KI can be used as a phase transfer catalyst. Finkelstein reaction, the reaction of converting brominated hydrocarbons into iodohydrocarbons with potassium iodide in DMAc. Specifically, the reagent potassium iodide is soluble in DMAc, but the potassium bromide produced by the reaction is insoluble and will precipitate out of the reaction solution, thereby promoting the continuous conversion of brominated hydrocarbons into iodohydrocarbons.

Compared with the above FT-IR spectras show in Figure 2, this FT-IR spectra of FAZO displays the characteristic hydroxyl stretches at 3420 cm$^{-1}$, and the C=C bond in the benzene ring stretches at 1595 cm$^{-1}$. In addition, the absorption peak at 1744 cm$^{-1}$ is caused by the C=O stretching vibration of the ester group, and the absorption peak at 1708 cm$^{-1}$ is caused by the C=C bond stretching vibration of the propylene group, indicating that 4-trifluoromethyl The acylation reaction of phenyl-4′-(9-hydroxynonoxy) azobenzene and methacryloyl chloride produces 9-(4-trifluoromethyl-4′-azophenol)-1-methacrylic acid nonyl ester.

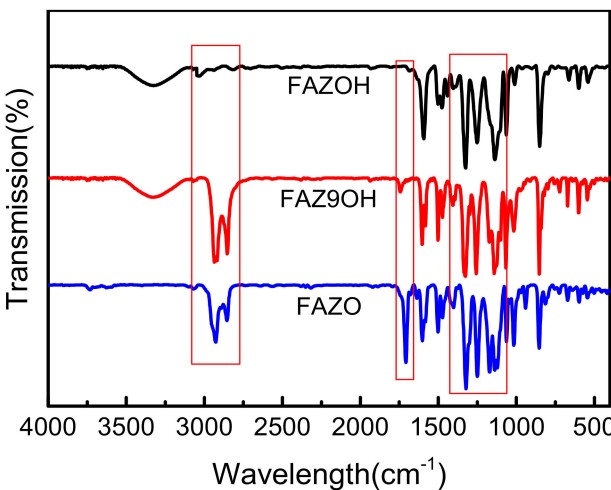

**Figure 2.** FT-IR spectra of 4-trifluoromethyl-4′-azophenol (FAZOH), 4-trifluoromethyl4′-azophenol etherified with 9-bromo-1-nonanol (FAZ9OH), and FAZO.

The $^1$H NMR of azo has been recorded from δ 0 to 9 ppm with CDCl$_3$ to confirm the azo formulation. Synthesized FAZOH was characterized by $^1$H NMR spectrum in Figure 3. we believe that the chemical shift of the protons of the benzene ring at positions a-d in the AZO domain can correspond to the chemical shift at 7.4 to 8.5 ppm in the figure, and the chemical shift at 7.88 ppm can be attributed to the phenolic hydroxyl group. The long alkyl chain is grafted onto azobenzene by etherification; the difference from the above $^1$H NMR spectra Figure 3 is that Figure 3 displays the characteristic long alkyl chain domain at e-m position, a chemical shift from 1.0–4.0 ppm. C=C group was modified onto FAZ9OH molecules, respectively, through acyl chlorination. The chemical information of the monomer was confirmed by the $^1$H NMR spectrum in Figure 3. In addition, $^1$H NMR spectroscopy confirmed that the FAZO monomer had been successfully synthesized. Specifically, Chemical shifts from 5.7 to 6.6 ppm were attributed to three H on the vinyl group at p, p′, and n positions, which indicated that vinyl was successfully modified onto AZO molecules.

The hydroxyl groups of the CNC were treated with 2-bromoisobutyryl bromide (BIBB) and then converted to a cellulose-based bromide carbonyl-bromide compound (CNC-Br) as the ATRP macroinitiator. (See the support information for more experimental details). It is worth mentioning that the success of surface modification was confirmed by FT-IR. IR spectra (Figure 4) of the unmodified CNC displays the characteristic hydroxyl group stretch at 3328 cm$^{-1}$, and the–C–O–C– bond in the structure stretch at 900–1100 cm$^{-1}$. By comparing with the IR spectrum of the unmodified CNC, it can be found that the IR spectrum of the brominated modified CNC has a peak at 1740 cm$^{-1}$, which is attributed to the twisted vibration of the C=O bond. It is believed that most grafted polymers exist on the surface of CNC because almost all of the hydroxyl groups of CNC are exposed on the surface of CNC, which facilitates the surface polymerization reaction. Soxhlet extraction was used to remove the unreacted reagents. FT-IR spectra of the CNC-PFAZO graft copolymer show the presence of asymmetrical and symmetrical CH$_2$ stretches at 2850 and 2930 cm$^{-1}$, and the stretching vibration absorption peak of the C=C bond in the benzene ring at 1595 cm$^{-1}$. The absorption peaks at 1325 cm$^{-1}$, 1257 cm$^{-1}$, 1140 cm$^{-1}$ and 1083 cm$^{-1}$ are in the trifluoromethyl group, respectively. The C–F bond's stretching vibration absorption peak, the absorption peak at 1744 cm$^{-1}$ is the C=O stretching vibration absorption peak in the ester group, and the absorption peak at 1708 cm$^{-1}$ is the C=C stretching vibration absorption peak in the propylene group. These all proved that it is grafted onto CNC.

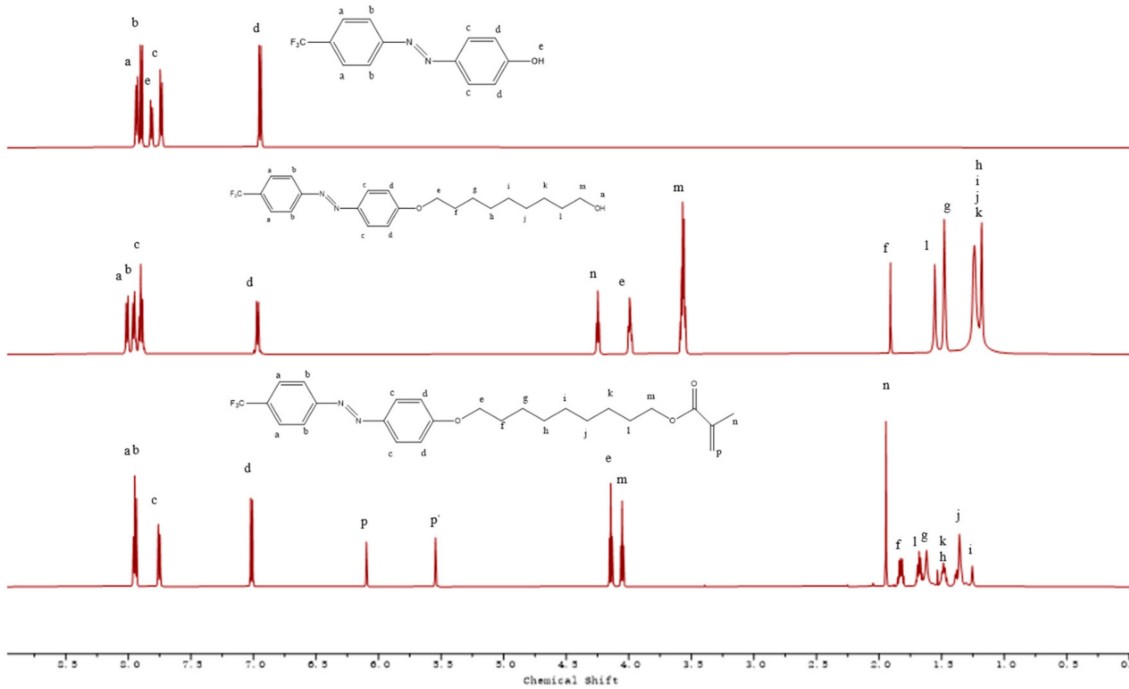

**Figure 3.** [1]H NMR spectra of FAZOH, FAZ9OH, and FAZO.

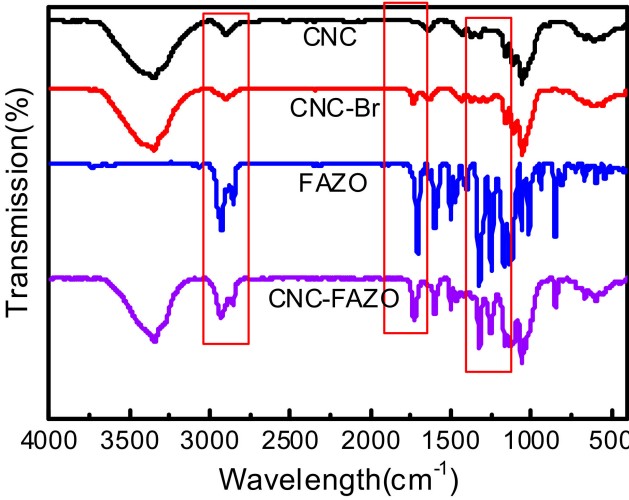

**Figure 4.** FT-IR spectra of cellulose nanocrystal (CNC), cellulose-based bromide carbonyl-bromide compound (CNC-Br), FAZO, and CNC-PFAZO.

The XRD patterns of the CNC, CNC-Br, and CNC-PFAZO are shown in Figure 5. CNC was grafted with the initiator 2-bromoisobutyryl bromide first, and then CNC-PFAZO was generated through a polymerization reaction. This reaction destroyed the transformation of hydrogen bonds in cellulose nanocrystals, resulting in a decrease in crystallinity. Additionally, as shown in Figure 6a–c, have obvious peaks at (2θ) 14.8°, 16.3°, 22.6°, and 34.0°, which are typical crystalline forms of cellulose I, consistent with the previous report by Fink et al. [23]

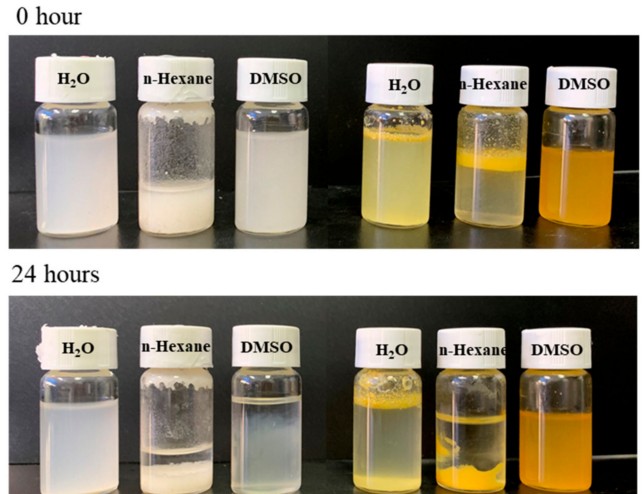

**Figure 5.** Photograph of two separate CNC samples: left: the CNC and right: the CNC-PFAZO.

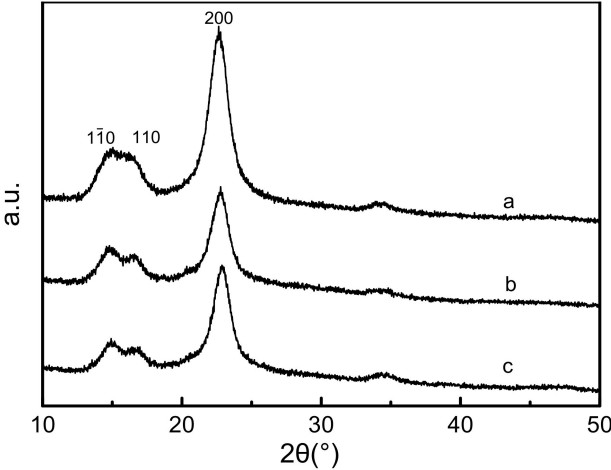

**Figure 6.** X-ray wide-angle scattering curves of (**a**) CNC, (**b**) CNC-Br, and (**c**) CNC-PFAZO.

In order to prove the solubility of CNC after grafting PFAZO, we dispersed the CNC and the CNC-PFAZO in different solvents (e.g., $H_2O$, n-hexane, DMSO), shows in Figure 6. It can be seen that at the beginning, CNC could be dispersed well in $H_2O$ and DMSO, but poorly dispersed in n-hexane, while the CNC-PFAZO was well dispersed in DMSO, appears orange and homogeneous, and almost no dispersion in water and n-hexane. The possible reason is that the surface of CNC after modification is highly hydrophobic, which hinders dispersion in water. After 24 h, it can be seen that the CNC partially precipitated in n-hexane and DMSO, and there was a layering phenomenon. After the modification, the CNC can also see obvious layering in n-hexane, but it is well dispersed in DMSO. This proved that the "grafting-from" procedure was successful.

In order to evaluate homogenous suspension, we characterized the hydrodynamic diameter of CNC before and after PFAZO. PFAZO is a hydrophobic polymer. Due to its surface contains polar groups, it is easy to agglomerate due to hydrogen bonding in polar solvents. Figure 7 shows the hydrodynamic diameter of the CNC aqueous solution is 123 nm and 2023 nm for CNC-PFAZO. It is clear that hydrodynamic diameter increases by increasing PFAZO length in block copolymer and also the addition of CNC into the DMSO. In organic solvents (like DMSO), this is an acceptable value.

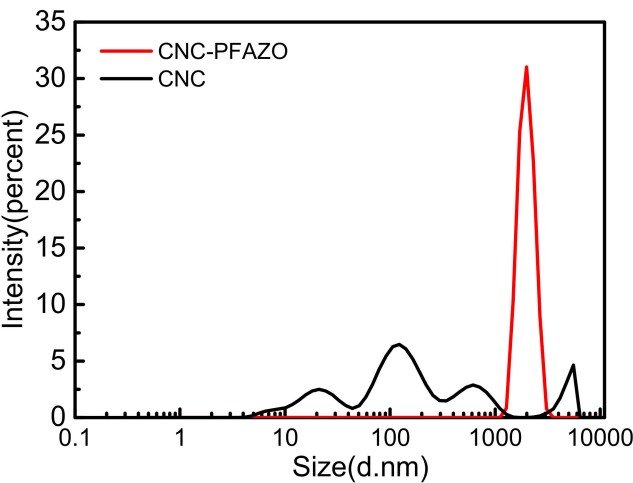

**Figure 7.** The hydrodynamic diameter of the CNC and the CNC-PFAZO by intensity.

The surface hydrophobicity of CNC-PFAZO is reflected by the value of the static contact angle. The water absorption speed of CNC film is so fast that it is impossible to measure the static contact angle. Surprisingly, Figure 8 shows that CNC-PFAZO film became hydrophobic with a contact angle of up to 134.4° (24 h). Even after 14 days, the contact angle remained at 132°. This verifies that hydrophilic CNC can be changed to hydrophobic by graft copolymerization modification due to the hydrophobic polymer chains covalently attached to the surface.

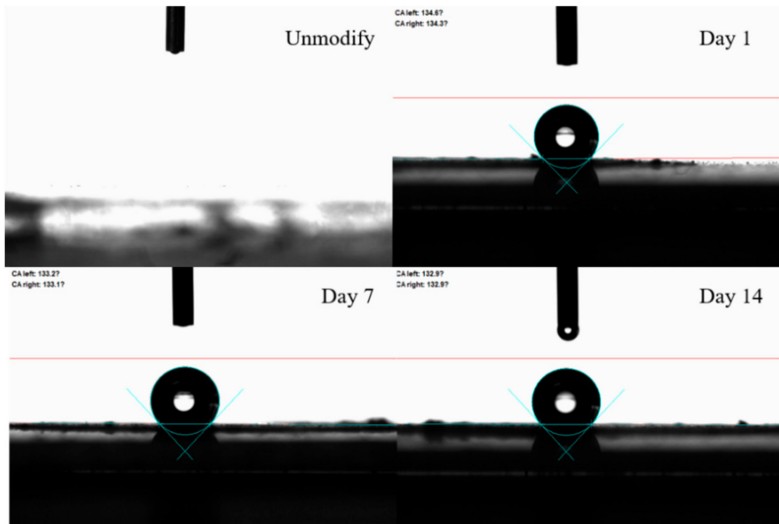

**Figure 8.** Contact angle of a water droplet on CNC and CNC-PAZO polymers ($H_2O$, 5 μL).

## 3. Conclusions

In conclusion, we used the residual hydroxyl groups on the CNC to react with 2-bromoisobutyryl bromide, and the initiator was therefore anchored on the CNC surface. Through atom transfer radical polymerization (ATRP), CNC was successfully grafting of azobenzene monomer. The CNC-PFAZO was highly hydrophobic with water contact angles of up to 134.4°, and it can remain highly hydrophobic property for more than a month or longer. This feature of CNC-PFAZO expands the application of CNC in hydrophobic substrates and lays the foundation for subsequent research on the behavior of optically responsive liquid crystals of cellulose-based azo polymers.

**Supplementary Materials:** The following are available online at http://www.mdpi.com/2504-477X/4/4/186/s1.

**Author Contributions:** Z.X. and S.P. contributed equally to this work. X.X. and G.Z. supervised the project. All authors revised the manuscript. All authors have read and agreed to the published version of the manuscript.

**Funding:** This work is supported by the National Natural Science Foundation of China (No. 51903094) and the Science and Technology Program of Guangzhou (No. 2019050001). The work is also partially supported by the National Key R&D Program of China (No. 2016YFB0401502), Program of Chang Jiang Scholars and Innovative Research Teams in Universities (No. IRT 17R40), Guangdong Provincial Laboratory of the Optical Information Materials and Technology (No. 2017B030301007), 111 projects and Yunnan expert workstation (2017IC011).

**Conflicts of Interest:** The authors declare no conflict of interest.

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
