# Peer review of "Highly Hydrophobic, Homogeneous Suspension and Resin by Graft Copolymerization Modification of Cellulose Nanocrystal (CNC)"

_jcs, doi:10.3390/jcs4040186_

Round 1

Reviewer 1 Report

The manuscript entitles "Highly Hydrophobic, Homogeneous and Colored Suspension and Resin by Graft Copolymerization Modification of Cellulose Nanocrystal (CNC)" can be beneficial to the nanocellulose research community. However, several issues should be properly addressed before publication-

  1. The authors should rewrite the title of the manuscript. They used the term "colored suspension" in the title, but they did not explain anything regarding the benefit of having color suspension in the manuscript.
  2. The authors must provide any experimental evidence before claiming homogenous suspension. They must provide experimental evidence like DLS/ turbidity data before claiming it.
  3. I am interested in their experimental set-up for their contact angle measurements. They provide contact angle data for 1-7 days (figure 7). If they measure data contentiously for 14 days, then how did they overcome the water evaporation problem? Moreover, surface morphology is also significantly controlling the contact angle of the substrate. The authors must provide surface roughness data along with the contact angle measurements.

Following reference should be included in the manuscript-

  1. Shrestha, Shikha, et al. "Surface hydrophobization of TEMPO-oxidized cellulose nanofibrils (CNFs) using a facile, aqueous modification process and its effect on properties of epoxy nanocomposites." Cellulose 26.18 (2019): 9631-9643.
  2. Chowdhury, Reaz A., et al. "High-Performance Waterborne Polyurethane Coating Based on a Blocked Isocyanate with Cellulose Nanocrystals (CNC) as the Polyol." ACS Applied Polymer Materials 2.2 (2019): 385-393.

Author Response

For details about the point-by-point response to the reviewer’s comments, please see the attachment, thanks

Reviewer 2 Report

Please refer to attached file for comments.

Author Response

For details about the point-by-point response to the reviewer’s comments, please see the attachment, thanks.

Round 2

Reviewer 2 Report

In my original review, I provided 3 pages of comments. The authors have elected to only respond to 6 points from my review. This is unacceptable and disrespects the peer review process.

Author Response

Please check the relevant information in the attachment, thanks.
